# A single exposure to altered auditory feedback causes observable sensorimotor adaptation in speech

Lana Hantzsch[1], Benjamin Parrell[1,2]*, Caroline A Niziolek[1,2]*

[1]Waisman Center, University of Wisconsin–Madison, Madison, United States; [2]Department of Communication Sciences and Disorders, University of Wisconsin–Madison, Madison, United States

**Abstract** Sensory errors induce two types of behavioral changes: rapid compensation within a movement and longer-term adaptation of subsequent movements. Although adaptation is hypothesized to occur whenever a sensory error is perceived (including after a single exposure to altered feedback), adaptation of articulatory movements in speech has only been observed after repeated exposure to auditory perturbations, questioning both current theories of speech sensorimotor adaptation and the universality of more general theories of adaptation. We measured single-exposure or 'one-shot' learning in a large dataset in which participants were exposed to intermittent, unpredictable perturbations of their speech acoustics. On unperturbed trials immediately following these perturbed trials, participants adjusted their speech to oppose the preceding shift, demonstrating that learning occurs even after a single exposure to auditory error. These results provide critical support for current theories of sensorimotor adaptation in speech and align speech more closely with learning in other motor domains.

*For correspondence:
bparrell@wisc.edu (BP);
cniziolek@wisc.edu (CAN)

**Competing interest:** The authors declare that no competing interests exist.

## Editor's evaluation

The paper establishes the presence of a single-trial adaptation response to the perturbation of the first formant of a vowel in speech production, an effect that should be of interest to the sensorimotor community in general. The analysis is conducted on existing data from 6 published studies and the effects are shown in a convincing fashion. The paper also explores the relationship between the within-trial compensation and the next-trial adaptation.

## Introduction

Auditory feedback plays a major role in both online execution and refinement of speech motor plans, as observed when the auditory feedback participants receive about their own speech is perturbed in real time (*Houde and Jordan, 1998*; *Purcell and Munhall, 2006b*; *Tourville et al., 2008*; *Villacorta et al., 2007*). Two types of behavior have been the primary focus of auditory perturbation studies in speech, which most typically alter a speaker's vowel formants (the resonant frequencies of the vocal tract that distinguish vowels). First, when unpredictable perturbations are delivered, speakers produce a *compensation* response—an online, within-trial adjustment to oppose the perturbation (*Purcell and Munhall, 2006b*; *Tourville et al., 2008*). Second, consistent perturbations lead to *sensorimotor adaptation*—a learned change in speech behavior that is observable from the onset of a subsequent speech movement and which persists even after the perturbation is removed (*Houde and Jordan, 1998*; *Purcell and Munhall, 2006a*).

These behaviors are widely considered to be driven by sensory prediction errors (differences between expected and perceived sensory feedback), although models differ in the proposed mechanism by which this occurs. In the DIVA (Directions Into Velocities of Articulators) model (*Tourville and Guenther, 2011*), sensory prediction errors lead to feedback-based corrective motor commands (i.e., the within-trial compensation response) which are subsequently incorporated into the feedforward motor program used for future productions of the same syllables, creating the adaptation response (*Kawato et al., 1987*). An alternative theoretical account of adaptation (*Houde and Nagarajan, 2011*) suggests sensory prediction errors instead directly lead to updates of internal models in the sensorimotor control system, either to forward models predicting the sensory outcomes of actions (*Bastian, 2006*; *Haith and Krakauer, 2013*; *Houde and Nagarajan, 2011*; *Krakauer and Mazzoni, 2011*; *Shadmehr et al., 2010*), to the control policy guiding action (*Hadjiosif et al., 2020*), or to both (*Wolpert et al., 1998*; *Wolpert and Kawato, 1998*).

Both the compensation-based and internal-model hypotheses of sensorimotor adaptation predict that learning in speech occurs progressively, with sensory feedback from each utterance causing updates to feedforward commands, such that changes in speech production should be evident even after a single trial with altered auditory feedback. Such *one-shot adaptation* has been observed in limb control, where a visuomotor perturbation on an isolated trial affects reach direction on the following trial (*Diedrichsen et al., 2005*; *Joiner et al., 2017*; *Ruttle, 2021*). However, the occurrence of such one-shot adaptation has not been definitively established in speech. Although *Cai et al., 2012*, observed that first formant (F1) production in the first 50 ms of perturbed trials which closely followed another perturbed trial tended to oppose the preceding perturbation's direction, more recent work explicitly testing for such single-trial effects did not find evidence of a measurable change (*Daliri et al., 2020*). This failure to find one-shot adaptation in speech questions both current theories of speech sensorimotor adaptation as well as the universality of domain-general theories (e.g., *Houde and Nagarajan, 2011*; *Kawato et al., 1987*; *Hadjiosif et al., 2020*).

Here, we aim to further investigate the mechanisms underlying sensorimotor adaptation by measuring one-shot adaptation in speech. To detect this potentially small effect, data from six prior studies (*Niziolek et al., 2014*; *Niziolek and Guenther, 2013*; *Niziolek and Parrell, 2021*; *Parrell et al., 2017*) were compiled for this analysis (131 total participants, 18–40 participants per study). In all studies, participants read aloud monosyllabic words while receiving real-time auditory playback of their speech. On a given trial, this feedback was either veridical (*unperturbed trials*) or unpredictably perturbed via an upward or downward shift in F1 (*perturbed trials*) (*Figure 1A*). Perturbed trials were used to calculate compensation responses, and unperturbed trials which occurred directly after a perturbed trial (*post-perturbation trials*) were used to calculate one-shot adaptation responses (see *Figure 1B*). We hypothesized that F1 values would be higher for trials that occurred directly after a

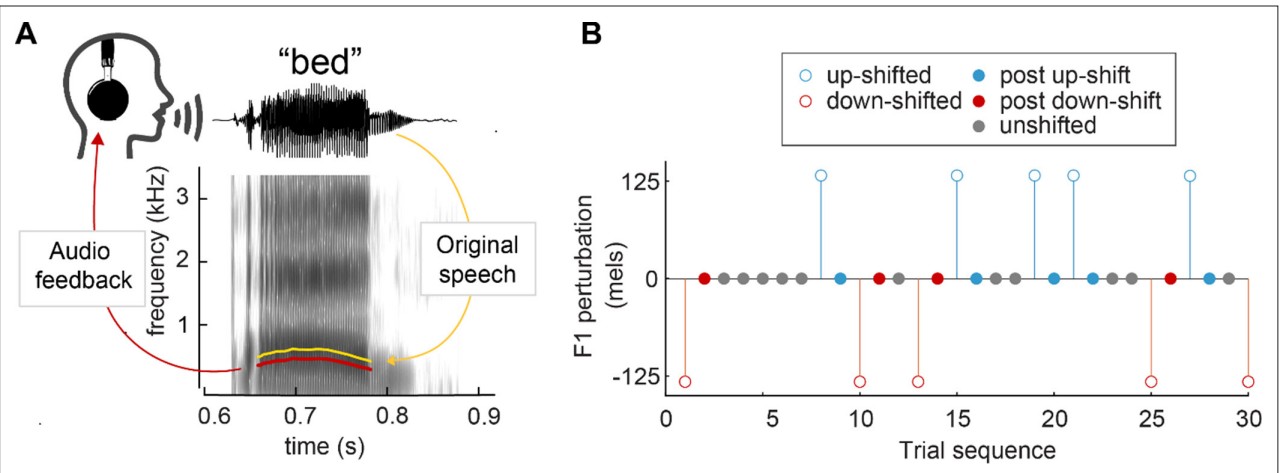

**Figure 1.** Perturbation methodology. (**A**) Spectrogram of the word 'bed', demonstrating an applied downward F1 perturbation. The F1 frequency of the audio feedback (red) is lowered from the original utterance (yellow). (**B**) Sample trial sequence from Study 4. Open circles indicate trials in which a perturbation was applied, used to calculate compensation. Closed circles indicate trials in which no perturbation occurred; 'post-up' and 'post-down' trials were used to calculate one-shot adaptation.

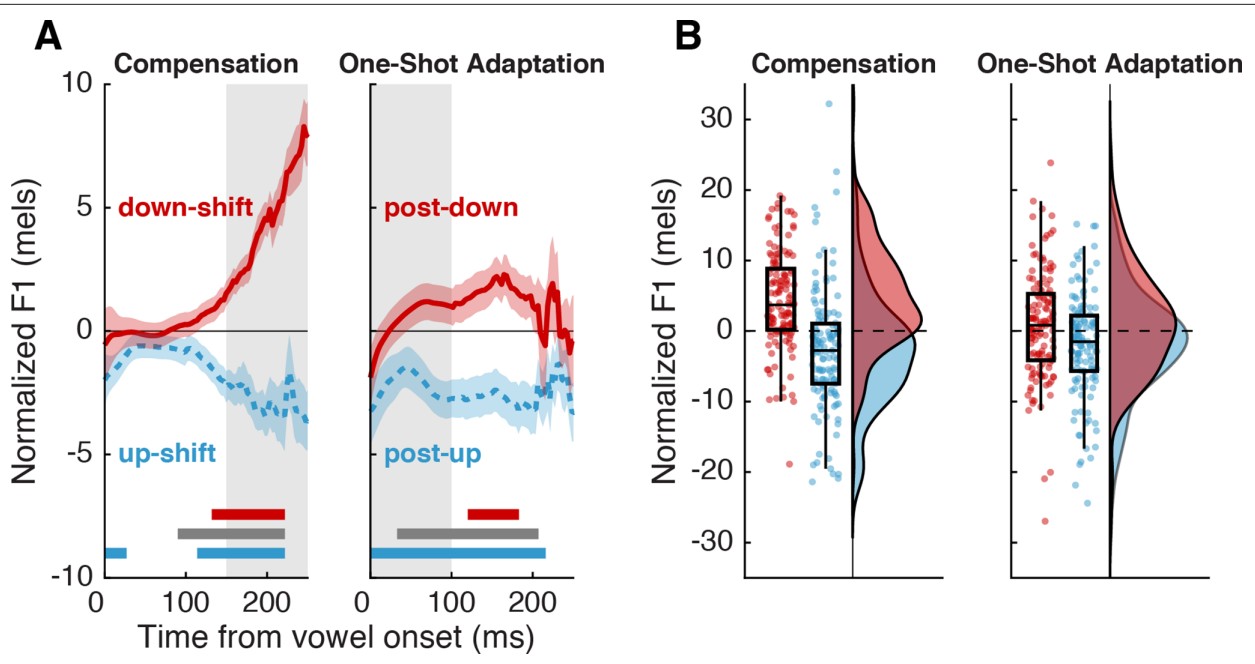

**Figure 2.** Behavioral responses to auditory perturbations. (**A**) Average normalized F1 for trials with upward (blue) or downward (red) perturbations. Error bars show standard error across participants. Highlighted regions illustrate the time periods of interest for compensation (left) and one-shot adaptation (right). Horizontal bars denote times with significant effects (p<0.05; n=131) as determined by cluster-based permutation tests (red and blue: difference from 0, gray: difference between conditions). (**B**) Probability distributions and boxplots of participants' average compensation and adaptation responses in the time periods of interest (n=131).

downward perturbation and lower in trials that occurred directly after an upward perturbation, such that they echo the preceding compensation response.

This approach also allows us to test the feedback-command-based hypothesis of adaptation in speech, which suggests that there should be a correlation between the magnitude of compensation and subsequent one-shot adaptation at the trial level. While this correlation has been observed in reaching (*Albert and Shadmehr, 2016*), most studies have failed to identify such a clear relationship in speech (*Daliri, 2021*; *Franken et al., 2019*; *Lester-Smith et al., 2020*; *Parrell et al., 2017*; *Raharjo et al., 2021*), possibly because they did not use such a direct trial-to-trial measurement method. The presence of such a relationship at the trial level would be compatible with both the feedback-command-based and internal-model hypothesis of adaptation; alternatively, the absence of such a relationship would only support the internal-model hypothesis.

## Results

### Compensation

In the 150–250 ms time window after vowel onset, trials in which an upward F1 shift occurred (*up-shifted trials*) had reliably lower F1 values (–3.99±0.33 mels (SE)) than trials in which a downward F1 shift occurred (2.69±0.33 mels) (*down-shifted trials*) ($\beta$=–6.93, SE=0.66, p<0.001, d=0.21; *Figure 2A*, left panel). This was also reflected at the individual level; participants' average F1 in the same time window was substantially lower across up-shifted trials (–2.75±1.56 mels) than down-shifted trials (4.40±1.25 mels) (paired t-test, *t*(130) = –7.00, p<0.001, d=0.91; *Figure 2B*, left panel). Normalized F1 was significantly different from 0 in both up-shifted trials (*t*(130) = –3.63, p<0.001, d=0.49) and down-shifted trials (*t*(130) = 7.24, p<0.0001, d=0.89). Additionally, a cluster-based permutation test showed significant differences from 0 starting at 100–125 ms after vowel onset for all trial types (*Figure 2A*, horizontal bars).

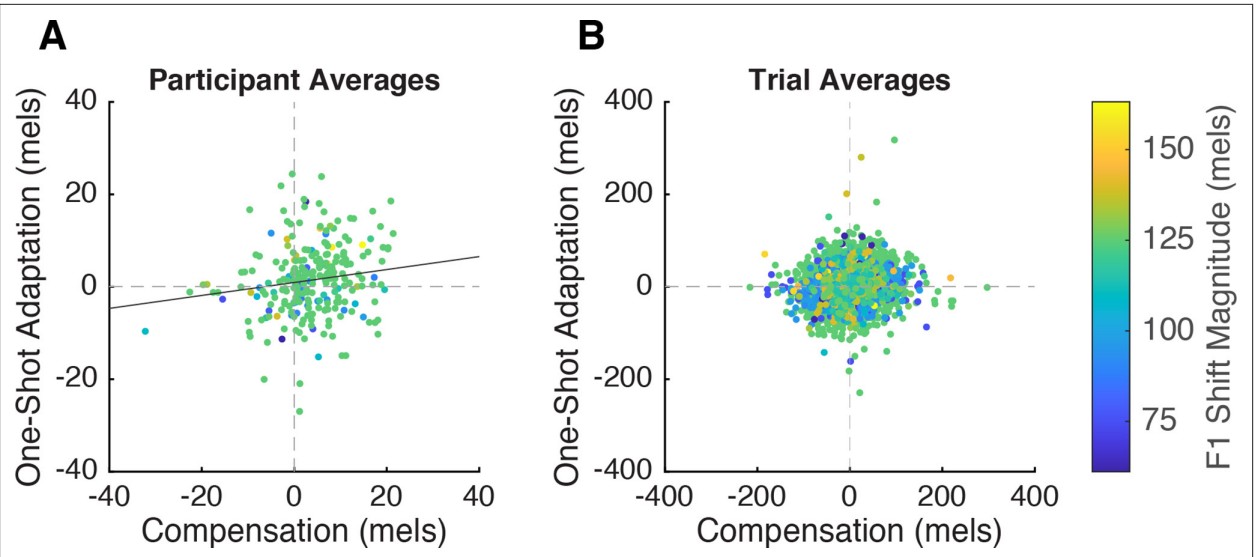

**Figure 3.** Correlation between compensation and one-shot adaptation. (**A**) Participant-level correlation. Each participant contributed two data points: their average response to up-shifted and their average response to down-shifted trials. The average applied F1 shift magnitude is displayed via the color gradient (blue = low shift magnitude, yellow = higher shift magnitude). The trend line (y=0.14x+0.93) represents the main effect of compensation on one-shot adaptation obtained from the linear mixed model. (**B**) Trial-level correlation. Each pair of perturbation and post-perturbation trials is a data point.

## One-shot adaptation

Participants produced one-shot adaptation responses which paralleled the directional pattern seen in the compensation response, though at a lower magnitude. In the 0–100 ms time window after vowel onset, trials that occurred immediately after an upward F1 shift (*post-up trials*) had F1 values (–1.55±0.26 mels) that were reliably lower than trials that occurred immediately after a downward F1 shift (*post-down trials*, 0.59±0.27 mels) ($\beta$=–2.14, SE=0.53, p<0.001, d=0.079; *Figure 2A*, right panel). Likewise, participants' average F1 was lower across post-up trials (–2.08±1.33 mels) than across post-down trials (0.82±1.39 mels) (paired t-test, $t(130) = -2.98$, p=0.0034, d=0.38; *Figure 2B*, right panel). Normalized F1 in post-up trials was significantly less than 0 in this time window ($t(130) = -3.2$, p=0.0016, d=0.38). While normalized F1 in post-down trials was numerically larger than 0, this difference was not significant in the 0–100 ms window ($t(130) = 1.2$, p=0.23, d=0.15); however, a cluster-based permutation test showed significant differences from 0 across the syllable for all trial types (*Figure 2A*, horizontal bars).

## Relationship between behavioral responses

At the participant level, there was a significant positive relationship between compensation and one-shot adaptation ($\beta$=0.14, SE=0.058, p=0.015, $\eta^2$=0.02), such that participants who produced larger compensation responses tended to adapt more (*Figure 3A*). Conversely, the trial-level model revealed no main effect of compensation response ($\beta$=–0.033, SE=–0.053, p=0.53) (*Figure 3B*). However, we did observe a small but significant interaction between shift magnitude and compensation response ($\beta$=0.16, SE=0.052, p=0.0023, $\eta^2$=0.0009), such that higher shift magnitudes elicited a stronger effect of compensation on adaptation. Along with the finding that larger shift magnitudes led to larger one-shot adaptation responses ($\beta$=7.46, SE=3.34, p=0.03, $\eta^2$=0.04), this suggests that trial-wise compensation may be predictive of adaptation only at larger shift magnitudes. Stronger evidence for a trial-level effect could be seen by correlating compensation and adaptation within each participant, which yielded a distribution of coefficients whose mean was significantly larger than 0 (mean $r$=0.21, 78/92 participants $r$>0, t=11.07, p<0.0001, d=1.6). There was no relationship between the strength of a participant's correlation between adaptation and compensation and their overall adaptation magnitude ($r$=–0.005, t=–0.04, p=0.964); in other words, it was not the case that this correlation was only observed in participants who adapted.

## Discussion

At both the trial and participant level, one-shot adaptation was detected in post-perturbation trials, where F1 values reliably opposed the perturbation in the previous trial. This shows that learning occurs continuously when the sensorimotor system detects a discrepancy between expected and perceived auditory feedback, as predicted by current models of sensorimotor adaptation in speech. While the magnitude of this one-shot adaptation may be small (1–2 mels), it is relatively substantial when accounting for the fact that a typical perturbation of ~100–150 mels causes an average F1 change of only 40–50 mels over the course of 100 or more trials (*Katseff et al., 2012*; *MacDonald et al., 2010*; *Munhall et al., 2009*; *Purcell and Munhall, 2006a*). Moreover, our estimate of one-shot adaptation is likely conservative for at least three reasons. First, all but one of the studies in our dataset presented multiple stimulus words in pseudorandom order; in ~53% of the trial pairs, participants pronounced different words on the perturbed and subsequent unperturbed trial. Although sensorimotor learning can generalize across words with the same vowel (*Rochet-Capellan et al., 2012*), such generalization is only partial, and a larger adaptation effect likely would have emerged with uniform word pairs. Second, our planned analysis evaluated adaptation during the first 100 ms of each vowel. Adaptation magnitude was greater (1.85±0.50 mels) during the 50–150 ms window, which excludes the consonant transition. Lastly, studies analyzed here used random, inconsistent perturbations across trials. Inconsistent perturbations are commonly used in studies of reaching adaptation in order to study learning that occurs after a single exposure (e.g., *Diedrichsen et al., 2005*; *Joiner et al., 2017*; *Ruttle, 2021*); however, such inconsistencies may decrease the rate of adaptation compared to consistent perturbations (*Albert et al., 2021*). One-shot adaptation in speech may therefore have an even greater magnitude in the typical case where perturbations are consistent across trials.

Though an individual's average compensation magnitude was reliably predictive of their average one-shot adaptation, a trial-level relationship was present but less reliable: it was mediated by shift magnitude and, when examined at the individual level, present in some but not all participants. It is unclear whether these two behavioral responses have a direct feedforward relationship (as predicted by the DIVA model), or if the observed correlations could best be explained by compensation and one-shot adaptation occurring via separate mechanisms driven by the same sensory error (as may be predicted by internal-model hypotheses). However, the less reliable within-participant relationship may be more consistent with models of adaptation that rely on updates to internal models compared to models that use feedback corrections to update future feedforward commands. Similarly mixed results on the causal relationship between feedback commands and feedforward learning have been reported in the reaching literature (*Albert and Shadmehr, 2016*; *Kim et al., 2021*; *Tseng et al., 2007*).

Overall, these results provide evidence that a single exposure to altered auditory feedback induces 'one-shot' adaptation in the speech sensorimotor system. This is consistent with current models of adaptation in speech specifically and in human movement more broadly; within these frameworks, one-shot adaptation is an effect that may continually build upon itself to create more enduring adaptation responses. Further comparison of single-trial vs. longer-timescale sensorimotor learning in the same individuals is warranted to strengthen this claim. The expected relationship between compensation and adaptation was observed reliably at the participant level and somewhat less reliably at the trial level. Our results provide evidence that adaptation in speech may operate in a similar manner as in other motor domains. As a well-learned natural behavior that relies primarily on implicit learning, speech offers a unique, ecologically valid paradigm to further our understanding of the underlying mechanisms driving sensorimotor adaptation.

## Materials and methods

### Participants

We reanalyzed data from six previous studies examining online compensation responses to formant frequency alterations with similar speech stimuli and perturbation schedules. Data were included if participants met inclusion criteria for their respective studies and if the formant shifts they received were opposite or near-opposite each other in 2D formant space (separated by an angle of 180±20° when plotted together in F1/F2 space). Data from 91 participants met these criteria; 40 of these participants contributed to two of the included studies. All participants were native speakers of American

**Table 1.** Summary of the included studies.

| | Study 1 (Parrell et al., 2017) | Study 2 (Parrell et al., 2021) | Study 3 (Niziolek and Parrell, 2021) | Study 4 (Niziolek and Parrell, 2021) | Study 5 (Niziolek and Guenther, 2013) | Study 6 (Niziolek et al., 2014) |
|---|---|---|---|---|---|---|
| # of participants included in analysis | 14/14 | 13/15 | 40/40* | 40/40* | 11/18 | 15/17 |
| # of outliers | 1 | 1 | 0 | 0 | 0 | 0 |
| Words | beck, bet, deck, debt, pet, tech | dead, fed, said, shed | bed, dead, head | bed, dead, head | bed, bet, dead, deb, debt, ped, tech, ted | head |
| # of trials | 160 | 120 | 240 | 240 | 400 | 800 |
| # of perturbed trials | 80 (50%) | 60 (50%) | 80 (33.33%) | 80 (33.33%) | 100 (25%) | 400 (50%) |
| F1 shift magnitude (mels) | 123.6±10 | 125 | 125 | 125 | 107.9±29.9 | 94.3±6.8 |
| Perturbation method | FUSP | Audapter | Audapter | Audapter | Audapter | FUSP |

The same group of participants contributed to both studies 3 and 4.

English and reported no history of speech, hearing, or neurological disorders. Informed consent and consent to publish was obtained for all participants. The experimental protocols were approved by the Institutional Review Board of the institutions from which data were collected: the University of Wisconsin–Madison, the Massachusetts Institute of Technology, the University of California, San Francisco, and the University of California, Berkeley. The University of Wisconsin–Madison Minimal Risk Research IRB approved our procedures to analyze the previously collected data (MRR IRB 2017-1509).

## Auditory perturbation

Details of the six studies are provided in *Table 1*. In all studies, participants spoke aloud monosyllabic English words containing the vowel /ɛ/ (as in *head*), which were presented as text on a screen. Simultaneously, participants heard real-time auditory feedback of their speech through headphones. On a pseudorandom subset of trials (25–50%), auditory feedback was altered with one of two real-time feedback perturbation systems, Audapter (*Cai et al., 2008*; *Tourville et al., 2013*) or Feedback Utility for Speech Production (FUSP) (*Katseff et al., 2012*; *Parrell et al., 2017*; *Figure 1*). Briefly, linear predictive coding (LPC) was used to model the vowel portion of the signal and apply a formant shift in real time during speech. Unaltered trials (50–75% of trials) underwent the same processing pipeline but with no alteration to the formants, such that auditory feedback in all trials had the same (minimal) delay. The magnitude and direction of the applied formant shift varied slightly across studies. Studies 1, 2, 3, and 4 shifted F1 upward and downward at a consistent magnitude (in mels or Hz) that was applied to all participants. Studies 5 and 6 each calculated participant-specific shift magnitudes for both F1 and F2 (in mels or Hz) along a vector pointing from the target vowel /ɛ/ to adjacent vowels /ɪ/ (as in *hid*) and /æ/ (as in *had*). For these studies, only the F1 portion of the vector was considered in the analysis; perturbations that increased F1 (/ɛ/ to /æ/) were considered 'up' shifts and perturbations that decreased F1 (/ɛ/ to /ɪ/) were considered 'down' shifts. All formant values were converted into mels for purposes of this analysis.

## Behavioral measures and statistical analysis

Our primary measure of interest was one-shot adaptation, an adaptive response that persists in the trial following an isolated perturbation. In order to examine whether one-shot adaptation is related to feedback-based corrections on the previous trial, we additionally measured the online compensation response. These behavioral responses were examined at both the trial level and the participant level.

Trials with a vowel duration of less than 100 ms were excluded from analysis (<1%). Two participants were excluded from the analysis as outliers (average compensation or one-shot response >4 SD from mean).

## Compensation

At the *trial level,* compensation response was operationalized as the mean normalized F1 produced during the 150–250 ms time window of trials in which a perturbation occurred (*perturbation trials*). More specifically, participant- and word-specific baseline F1 trajectories were first calculated from the F1 trajectories of unperturbed trials (*baseline trials*). The F1 trajectory of each perturbation trial was then normalized by subtracting the word-specific baseline mean F1 trajectory from it. The compensation response for each perturbation trial was then defined as the mean F1 value within 150–250 ms after vowel onset, after the typical onset latency of compensation. A 200–300 ms time window was originally planned for this analysis; however, only 46% of produced vowels had a duration of at least 300 ms, whereas 80% of vowels lasted until the end of the 150–250 ms time window.

Average compensation response was also calculated at the *participant level*, operationalized as a participant's mean normalized F1 across the 150–250 ms window of their perturbation trials. Again, the F1 trajectory of each perturbation trial was normalized via a participant- and word-specific baseline. Then for each participant, two average F1 trajectories were calculated: one trajectory that averaged the normalized trajectories across all trials containing an upward perturbation and one trajectory that averaged across all trials containing a downward perturbation. The participant's average compensation response for each perturbation direction (up and down) was calculated as the mean F1 value in the 150–250 ms time window after vowel onset of these averaged perturbation trajectories.

In the *trial level* analysis, a linear mixed model was employed to investigate the effect of perturbation direction on compensation response: *Compensation response ~ perturbation direction + (1 | participant) + (1 | study)*. Effect size was calculated by dividing $\beta$ by the residual standard deviation. At the *participant level*, a paired t-test was used to evaluate the distribution of participants' mean compensation response to upward perturbations vs. downward perturbations. Additional one-sample t-tests were conducted for each perturbation type against a mean of 0. Cohen's *d* was calculated to determine effect size. Examining the entire 0–250 ms window, a cluster-based permutation test was used to find clusters of time points in which the compensation response for each condition differed from 0 and, separately, from each other (*Maris and Oostenveld, 2007*).

## One-shot adaptation

At the *trial level*, one-shot adaptation response was calculated as the mean normalized F1 produced in the first 100 ms of unperturbed trials that occurred directly after a perturbed trial (*post-perturbation trials*). Again, participant- and word-specific baseline trajectories were calculated, though using F1 trajectories from unperturbed trials that directly followed another unperturbed trial (*baseline trials*). The F1 trajectories of each post-perturbation trial were then normalized by subtracting the word-specific baseline mean F1 trajectory. The one-shot adaptation response for each post-perturbation trial was calculated as the mean F1 value in the first 100 ms of the normalized trajectory. Only F1 values from the initial 100 ms of the vowel were included, limiting the influence of auditory-based feedback control mechanisms, which have a latency of 100–150 ms in speech (*Cai et al., 2012*; *Parrell et al., 2017*; *Tourville et al., 2008*).

At the *participant level*, the one-shot adaptation response was calculated as a participant's mean normalized F1 in the first 100 ms of their average post-perturbation trial F1 trajectory. Again, the F1 trajectory of each post-perturbation trial was normalized via a participant- and word-specific baseline. Then, for each participant, two average F1 trajectories were calculated: one trajectory that averaged the normalized trajectories across all trials that occurred after an upward perturbation and one trajectory that averaged across all trials that occurred after a downward perturbation. The participant's average one-shot adaptation response for each perturbation direction (up and down) was calculated as the mean F1 value in the first 100 ms of these averaged post-perturbation trajectories.

At the *trial level*, a linear mixed model was employed to investigate the effect of perturbation direction on one-shot adaptation: *One-shot adaptation response ~ perturbation direction + (1 | participant) + (1 | study)*. Effect size was calculated by dividing $\beta$ by the residual standard deviation. At the *participant level*, a paired t-test was implemented to assess the distribution of participants' mean one-shot adaptation response to upward perturbations vs. downward perturbations. Additional one-sample t-tests were conducted for each post-perturbation type against a mean of 0. Cohen's *d* was calculated to determine effect size and conduct a power estimation. As in the compensation analysis,

a cluster-based permutation test identified clusters of time points in which the adaptation response for each condition differed from 0 and, separately, from each other.

### Relationship between behavioral responses

In order to assess the relationship between compensation and the one-shot adaptation that followed it, we fitted a linear mixed effects model to one-shot adaptation with compensation, perturbation magnitude, and perturbation condition as fixed factors and with participant as a random intercept. To maintain a standardized magnitude measure between the two perturbation directions, compensation and one-shot adaptation responses from upward-shifted trials were multiplied by –1, removing the directional difference between up- and down-perturbation conditions. Separate analyses were conducted at the participant level (averaging across all trials) and at the individual trial level. To avoid problems in the linear models caused by predictors of very different scales, each perturbation magnitude was normalized by dividing by the mean of all perturbation magnitudes across participants. Study was not included as a separate random intercept in the model as it introduced singularity to the model due to its collinearity with participant and shift magnitude.

At the trial level, compensation response was intended to be included as a random slope by participant, however, this slope was removed because the model failed to converge. As a separate test of the within-subject relationship between compensation and one-shot adaptation, a one-sample t-test was conducted on participant correlations between their compensation and adaptation responses (sign-corrected). Pearson's $r$ was calculated for each participant, correlating compensation with subsequent adaptation at the trial level. A Fisher transformation was used to convert the correlation coefficients into $z$-scores prior to running the one-sample t-test; the mean was converted back to an $r$ value for interpretation here.

All statistical analysis was conducted in R (*R Development Core Team, 2020*). Linear mixed effects models and their simplest explanatory models (calculated via stepwise regression) were generated using the *lme4* package (*Bates et al., 2015*). Statistical significance of the final model was assessed with the *lmerTest* package, which uses the Satterthwaite method to estimate degrees of freedom (*Kuznetsova et al., 2017*). Power analyses for t-tests were conducted with the *pwr* package (*Champely, 2020*). Correlation between compensation and one-shot adaptation was then assessed with a Pearson's $r$ correlation coefficient using the *MuMIn* package (*Barton, 2020*). Effect sizes were calculated using the *effectsize* package (*Ben-Shachar et al., 2020*). Data and analysis code is available at https://github.com/blab-lab/postMan, (*Hantzsch, 2022a* copy archived at swh:1:rev:6cf539d0662552f-27d1560a250285e49edde82c4). Some of the functions rely on additional code available at https://github.com/carrien/free-speech, (*Hantzsch, 2022b* copy archived at swh:1:rev:e065de8fa8c49ac9795f1865df5d171f0869666a).

## Acknowledgements

This work was funded by National Institutes of Health grants R01 DC017091 (BP) and R00 DC014520 (CAN).

## Additional information

### Funding

| Funder | Grant reference number | Author |
| --- | --- | --- |
| National Institutes of Health | DC014520 | Caroline A Niziolek |
| National Institutes of Health | DC017091 | Benjamin Parrell |

The funders had no role in study design, data collection and interpretation, or the decision to submit the work for publication.

## Author contributions

Lana Hantzsch, Data curation, Formal analysis, Methodology, Software, Visualization, Writing – original draft, Writing – review and editing; Benjamin Parrell, Conceptualization, Data curation, Formal analysis, Funding acquisition, Investigation, Methodology, Project administration, Software, Supervision, Visualization, Writing – review and editing; Caroline A Niziolek, Software, Conceptualization, Data curation, Formal analysis, Supervision, Visualization, Writing – review and editing, Funding acquisition, Investigation, Writing – original draft, Project administration

## Author ORCIDs

Benjamin Parrell http://orcid.org/0000-0003-2610-2884
Caroline A Niziolek http://orcid.org/0000-0002-6085-1371

## Ethics

Informed consent and consent to publish was obtained for all participants. The experimental protocols were approved by the Institutional Review Board of the institutions from which data were collected: the University of Wisconsin-Madison, the Massachusetts Institute of Technology, the University of California, San Francisco, and the University of California, Berkeley. The University of Wisconsin-Madison Minimal Risk Research IRB approved our procedures to analyze the previously collected data (MRR IRB 2017-1509).

## Decision letter and Author response

Decision letter https://doi.org/10.7554/eLife.73694.sa1
Author response https://doi.org/10.7554/eLife.73694.sa2

## Additional files

### Supplementary files

• Transparent reporting form

### Data availability

Data and analysis code are available on GitHub at https://github.com/blab-lab/postMan, (copy archived at swh:1:rev:6cf539d0662552f27d1560a250285e49edde82c4).

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
