## [Editor Report]

The paper establishes the presence of a single-trial adaptation response to the perturbation of the first formant of a vowel in speech production, an effect that should be of interest to the sensorimotor community in general. The analysis is conducted on existing data from 6 published studies and the effects are shown in a convincing fashion. The paper also explores the relationship between the within-trial compensation and the next-trial adaptation.

---

## [Decision Letter]

**Decision letter after peer review:**

Thank you for submitting your article "A single exposure to altered auditory feedback causes observable sensorimotor adaptation in speech" for consideration by *eLife*. Your article has been reviewed by 2 peer reviewers, including Jörn Diedrichsen as Reviewing Editor and Reviewer #1, and the evaluation has been overseen by Barbara Shinn-Cunningham as the Senior Editor.

Essential revisions:

1) The relationship between online correction and subsequent adaptation across perturbation sizes need to be clarified – especially the between and within-person relationships need to be more cleanly separated – see detailed comments by both reviewers.

2) The adaptation response in each direction (up and down perturbation) should be tested against zero.

3) The authors should discuss to what degree adaptation to random perturbations can serve as a good model of adaptation response occurring during systematic perturbations. For reaching there is an extensive literature on potential differences and similarities between random and systematic perturbations.

*Reviewer #1 (Recommendations for the authors):*

Line 143: Was there a difference between same-word and different-word adaptation in this study?

Line 277-288: It would be good to clarify both in the results and here in the beginning of the paragraph that the compensation, one-shot adaptation response, and perturbation are entered in the model corrected for perturbation direction. This information comes a bit late.

Line 277-288: Was perturbation size included in the model as a continuous variable, or in a one-hot categorical encoding? It seems that the range was 96-125mels across study with one size per study? When you report a significant interaction between perturbation size and compensatory response - what shape did the interaction take? It would be good to show this result broken up by study, so the reader can judge to what degree this maybe drive by differences between studies.

Line 289ff: Using Monte-carlo study to test for within-subject effects is somewhat indirect. Why not use the following simpler and (in my opinion) more convincing analysis: Estimate the slope of the compensation-adaptation for each subject separately? Then one can test is one-sample t-test of all the slopes against zero - or test for differences across perturbation sizes.

*Reviewer #2 (Recommendations for the authors):*

L 183 What does perturbations being separated by 180 deg in F1/F2 space mean? Please explain.

L 101 Responses to upward and downward shifts are reported to be statistically different than one another. However, it would be important to show that each shift is reliably different than zero. Is this the case? {plus minus} values are given. Are these standard errors, and if not, what are they? Standard errors would be helpful.

Figure 2. What is indicated by the error bars in the time-series plots?

L 123 The interaction effect that is described starting on L 123 needs to be better explained. Ditto for the claim that a Monte Carlo simulation suggests this is not a between subjects effect.

---

## [Author Response]

Essential revisions:1) The relationship between online correction and subsequent adaptation across perturbation sizes need to be clarified – especially the between and within-person relationships need to be more cleanly separated – see detailed comments by both reviewers.

We have clarified the results of our trial-level linear model, explaining the effect of perturbation size on the relationship between online compensation and subsequent adaptation (lines 130-131). We have also followed the advice of Reviewer 1, replacing our Monte Carlo simulation analysis with an evaluation of within-person trial-wise correlations between compensation and adaptation. This analysis did yield evidence that these correlation coefficients were reliably larger than zero across participants, indicating a relationship at the trial level. We have amended our discussion to address this finding (lines 166-177).

2) The adaptation response in each direction (up and down perturbation) should be tested against zero.

We have added this analysis: the adaptation response was significantly different from 0 in the pre-defined time window of interest for trials following upward perturbations (post-up). The response was numerically but not significantly larger than 0 in this time window for trials following downward perturbations (post-down); however, a cluster-based permutation analysis that considered all time points across the syllable showed significant differences from 0 in both post-up and post-down conditions (see horizontal bars on Figure 2A). We have added this result to lines 120-122 and describe the method in lines 263-266 and 295-297.

3) The authors should discuss to what degree adaptation to random perturbations can serve as a good model of adaptation response occurring during systematic perturbations. For reaching there is an extensive literature on potential differences and similarities between random and systematic perturbations.

We have added a brief discussion of this point while attempting to remain within word limits. Specifically, the first paragraph of the discussion has been expanded to discuss how random perturbations very similar to those used here have been used extensively in the reaching literature studying one-shot learning (e.g., Diedrichsen et al., 2005; Joiner et al., 2017; Ruttle et al., 2021), and how inconsistent perturbations have recently been shown to decrease adaptation (Albert et al., 2021). If anything, this suggests that we may be underestimating the magnitude of the adaptation response to more consistent perturbations. As the aim of our study was specifically to demonstrate the existence of adaptation following a single exposure to altered auditory feedback, we feel that this is not a major issue.

Reviewer #1 (Recommendations for the authors):Line 143: Was there a difference between same-word and different-word adaptation in this study?

Although the proportion of perturbed trials followed by the same vs. different words is close to 50% on average, this proportion is very different across studies, ranging from less than 15% to 100%; in other words, this comparison would, in effect, compare trials from studies with high word overlap with trials from studies with low word overlap (rather than comparing the two types of trials within individuals within a given study). Given that studies varied in many parameters including shift magnitude and proportion of shifted trials, unfortunately, we do not think the datasets used here are appropriate for the proposed analysis.

Line 277-288: It would be good to clarify both in the results and here in the beginning of the paragraph that the compensation, one-shot adaptation response, and perturbation are entered in the model corrected for perturbation direction. This information comes a bit late.

We have moved this information earlier in this paragraph and in the results.

Line 277-288: Was perturbation size included in the model as a continuous variable, or in a one-hot categorical encoding? It seems that the range was 96-125mels across study with one size per study? When you report a significant interaction between perturbation size and compensatory response - what shape did the interaction take? It would be good to show this result broken up by study, so the reader can judge to what degree this maybe drive by differences between studies.

Perturbation size was included as a continuous variable, as it was not always constant within a study. The interaction between perturbation magnitude and compensation was such that a stronger perturbation magnitude was related to a stronger effect of compensation on one-shot adaptation (now clarified on lines 130-131).

Line 289ff: Using Monte-carlo study to test for within-subject effects is somewhat indirect. Why not use the following simpler and (in my opinion) more convincing analysis: Estimate the slope of the compensation-adaptation for each subject separately? Then one can test is one-sample t-test of all the slopes against zero - or test for differences across perturbation sizes.

We thank the reviewer for the suggestion; we have replaced the Monte Carlo simulation with a one-sample t-test of participant compensation-adaptation correlations. The correlation coefficients were converted to z-scores via a Fischer transform prior to running the one-sample t-test. This analysis did yield a distribution of correlation coefficients that was significantly greater than 0, lending more evidence to a trial-level relationship; this is now explained and discussed in lines 133-139 & 166-177.

Reviewer #2 (Recommendations for the authors):L 183 What does perturbations being separated by 180 deg in F1/F2 space mean? Please explain.

Some studies included perturbations in both the first and second formant (F1 and F2). These perturbations can be conceptualized as a vector, using the F1 perturbation on the x-axis and the F2 perturbation on the y-axis (i.e., plotting in F1/F2 space). Thus, if two perturbations are exactly opposite of each other, the two vectors should be 180 degrees from each other in this plot, pointing in opposite directions. We have clarified this point on line 198.

L 101 Responses to upward and downward shifts are reported to be statistically different than one another. However, it would be important to show that each shift is reliably different than zero. Is this the case?

We now report cluster-based permutation tests to show differences from 0 for all conditions (see explanation above).

{plus minus} values are given. Are these standard errors, and if not, what are they? Standard errors would be helpful.

We have converted these values to standard error (originally standard deviation), and specified this on line 102.

Figure 2. What is indicated by the error bars in the time-series plots?

The error bars indicate standard error across participants. This is now specified in the figure caption.

L 123 The interaction effect that is described starting on L 123 needs to be better explained. Ditto for the claim that a Monte Carlo simulation suggests this is not a between subjects effect.

We clarify the interaction effect, which showed that the relationship between compensation and one-shot adaptation was mediated by shift magnitude: greater shift magnitudes elicited a stronger effect of compensation at the trial level. We have replaced our Monte Carlo simulation with an analysis of within-participant correlation coefficients, as suggested by Reviewer 1, and as described above, now include a discussion of the finding that the distribution of these coefficients is significantly greater than 0, lending additional evidence for a trial-level relationship in the majority of participants.